# Health-Related Quality of Life and Its Socio-Demographic and Behavioural Correlates during the COVID-19 Pandemic in Estonia

**DOI:** 10.3390/ijerph19159060

**Published:** 2022-07-25

**Authors:** Merili Tamson, Rainer Reile, Diana Sokurova, Kaire Innos, Eha Nurk, Kaia Laidra, Sigrid Vorobjov

**Affiliations:** 1National Institute for Health Development, 11619 Tallinn, Estonia; rainer.reile@tai.ee (R.R.); diana.sokurova@tai.ee (D.S.); kaire.innos@tai.ee (K.I.); eha.nurk@tai.ee (E.N.); kaia.laidra@tai.ee (K.L.); sigrid.vorobjov@tai.ee (S.V.); 2Institute of Psychology, University of Tartu, 50409 Tartu, Estonia; 3Institute of Family Medicine and Public Health, University of Tartu, 50411 Tartu, Estonia; 4Health Research Institute, Lithuanian University of Health Sciences, 47181 Kaunas, Lithuania

**Keywords:** health-related quality of life, EQ-5D, health behaviour, COVID-19 pandemic, Estonia, inequalities

## Abstract

The aim of the study was to analyse health-related quality of life (HRQoL) among the Estonian general population and its socio-demographic and behavioural correlates during the COVID-19 pandemic. Longitudinal data on 1781 individuals from an Estonian rapid-assessment survey on COVID-19 were used. HRQoL was assessed with the EQ-5D-3L in June 2020 (baseline) and in May 2021 (follow-up). The HRQoL index score and its socio-demographic and behavioural variations were analysed using paired *t*-tests and Tobit regression modelling. Statistically significant declines in mean EQ-5D index scores were observed for all socio-demographic and behavioural variables considered. Most of these changes were due to increased reporting of problems in the pain/discomfort and anxiety/depression health domains. Older age, being unemployed or economically non-active and having financial difficulties were significantly associated with lower HRQoL in both baseline and follow-up measurements. In the follow-up data, women had significantly lower HRQoL compared to men, whereas higher education proved to be the only protective factor regarding HRQoL. Unhealthy dietary habits and low physical activity had a negative impact on the HRQoL score in the follow-up data. These results indicate that the COVID-19 pandemic has had a substantial impact on HRQoL in the Estonian population.

## 1. Introduction

The ongoing coronavirus disease (COVID-19) pandemic has had a drastic impact on population health outcomes due to increased morbidity and mortality. Two and a half years since the first cases were identified in China [1], over 552 million COVID-19 infections have been registered and over 6.4 million deaths confirmed globally as of July 2022 [2]. As a result, estimated global life expectancy has reduced by more than 1.6 years—a change that is unprecedented in recent history [3].

The pandemic has disrupted the daily lives of the majority of people. In addition to the direct health effects due to COVID-19 infection, the potential indirect pathways to deteriorating health include changes in employment or work practices [4], loss or reduction in income [5] and limitations in accessing education [6] or healthcare and social services [7]. Pandemic-related restrictions have affected health-related behaviours [8] and social relationships [9], leading to an increase in mental health complaints during the pandemic [10,11,12].

While these are just a few mechanisms that explain the pandemic’s health implications, they provide an extensive yet relevant list of indicators that could be used to capture these effects. Generic health-related quality of life (HRQoL) instruments offer an analytic solution to this problem as they provide a multidimensional summary of health status that can be compared across a variety of diseases/conditions and used for different populations [13]. One of the most used HRQoL instruments is the EQ-5D by EuroQol Group [14], which covers five health domains (mobility, self-care, daily activities, pain/discomfort, anxiety/depression). Most of the available evidence on HRQoL during the pandemic covers COVID-19 patients and indicates a variety of clinical manifestations (most notably a reduction in physical health and an increase in mental health complaints), even several weeks after hospital discharge [15,16]. Previous evidence on longitudinal HRQoL data among the general population in European countries, including from the EQ-5D generic instrument, is relatively scarce. Available studies are mostly based on cross-sectional data and either use sample comparisons between infected and healthy subgroups [17] or refer to earlier population norms [18]. However, the results of the few published longitudinal studies from Denmark [19] and Japan [20] demonstrate individual-level declines in both mental and physical health domains during the pandemic.

This study contributes to the field by providing comparable longitudinal data on HRQoL during the pandemic in Estonia. The first COVID-19 cases were registered in Estonia (population 1.3 million) in late February 2020, with the 14-day incidence rate during the first wave reaching 56.6 cases per 100,000 in April 2020. Its epidemiological impact was modest compared to the second and third wave of the pandemic when the incidence rate peaked at >1500 cases per 100,000 (Figure 1) and the prevalence of SARS-CoV-2 RNA positive tests reached 2.7% [21]. Although pandemic-related health effects among the general population have so far been described as an increase in perceived stress [22] and mortality rates [23], data on potential impact on HRQoL have not yet been published.

Consequently, the overall aim of the paper is to study HRQoL among the Estonian general population and to analyse socio-demographic and behavioural correlates during the COVID-19 pandemic. More specifically, we will focus on the following research questions: (a) which HRQoL scores are reported at two points in time ten months apart during the pandemic in Estonia? (b) which socio-demographic and health behaviour indicators had an impact on HRQoL score at the individual level and did this pattern change during the follow-up?

## 2. Materials and Methods

### 2.1. Data

This study uses longitudinal data from an Estonian rapid-assessment survey on COVID-19 [25]. The survey was conducted as a repeated web survey with three waves between April 2020 and May 2021. The nationally representative stratified random sample (by sex and age group) drawn from the population registry included 12,000 individuals aged 18–79 years with a valid e-mail address. After the first anonymized cross-sectional survey (*n* = 4606; adjusted response rate 40.3%), the survey design was altered to longitudinal with individually linkable data collections planned for two additional waves. As HRQoL was included in the survey starting from the second wave, this paper is based on the data from the second and third waves of the survey.

The second survey wave (hereinafter baseline; *n* = 3464; adjusted response rate 31.1%) was carried out from 11 June to 20 July 2020 (Figure 1). It was timed in accordance with the epidemiological situation after the end of the first pandemic wave; infection and hospitalization rates had declined, and most restrictions had been lifted due to the ending of lockdown in the previous month. At this time (vaccination was not yet available), quarantine for those with COVID-19 infection and their close contacts existed, social distancing as well as capacity limitations for premises were mandatory, restrictions on alcohol sales and crossing the Estonian border were in place and public events were banned until 30 June.

The third survey wave (hereinafter follow-up; *n* = 3604; adjusted response rate 34.6%) was conducted from 13 April to 5 May 2021 and coincided with the second pandemic wave in Estonia (Figure 1). Just one month before the third survey wave, Estonia had the highest infection rates in the world. During the survey period, strict restrictions existed for the first two weeks, e.g., mask wearing was required, all schools applied distance learning, most shops were closed, capacity restrictions for premises and requirements for social distancing were in place. Restrictions on outdoor and indoor sports as well as on hobby education were in force until 26 April. From 3 May, primary schools and pupils with special needs were allowed contact learning, shops and museums opened, outdoor catering was allowed and limitations on sporting events were eased. Vaccination was available for certain occupational groups (e.g., medical and nursing personnel, teachers).

From the baseline survey, 61.1% (*n* = 2116) of respondents also participated in the follow-up survey. Of those who participated in both the baseline and the follow-up, HRQoL questions for measuring the EQ-5D score were answered by 1824 unique respondents. However, 21 respondents answered the HRQoL questions in the baseline but not in the follow-up, and 22 respondents did not answer the HRQoL questions in the baseline but did in the follow-up. Therefore, 43 unique respondents were excluded from the analytic sample to avoid individual variation in comparing results from two time points. In this study, only those respondents who answered the HRQoL questions in both survey waves (*n* = 1781; 51.4% of the second survey wave’s total sample and 49.4% of the third survey wave’s total sample; 84.1% of the longitudinal analysis sample) were included in the analysis.

### 2.2. Variables

HRQoL, measured using EuroQol’s EQ-5D-3L [26] descriptive system, was the dependent variable in the analysis. The instrument evaluates health status in five domains (mobility, self-care, usual activities, pain/discomfort and anxiety/depression) using a three-point scale: (1) no problems; (2) some problems; and (3) extreme problems. The domain scores are then combined into up to 243 health states (e.g., 11,213), which can be converted into a weighted health state index. For the latter, the UK population time trade-off tariffs ranging from 1 (for the best state, 11,111) to −0.594 (for the worst state, 33,333) [27,28] were used in this study.

Respondents’ demographic background was described by sex, age and ethnicity and socio-economic background by education, employment and households’ financial situation. A common dichotomous classification of sex—male and female—was used. Age was grouped into five categories: (1) 18–29; (2) 30–44; (3) 45–54; (4) 55–64; and (5) ≥65-years. Self-reported ethnicity was grouped as: (1) Estonians; and (2) other ethnic groups. Educational level refers to the highest level of education obtained and was aggregated into three groups: (1) primary or lower; (2) secondary or vocational; and (3) tertiary or higher education. Current employment status was reported in both baseline and follow-up and divided into categories: (1) employed; and (2) unemployed/non-active, referring to various economically non-active groups. Similarly, financial status was reported twice; it captured a subjective assessment of a household’s financial wellbeing during the past month and was categorized as: (1) comfortable; (2) sufficient; and (3) difficulty coping.

Health behaviour indicators in the analysis included self-estimated alcohol consumption, smoking status, physical activity and dietary habits. The frequency of alcohol consumption during the past 12 months was aggregated into a binary variable: (1) consumed alcohol ≥ 4 times a week (high risk consumption); and (2) consumed alcohol ≤ 3 times a week (low risk consumption). The indicator of smoking status included both traditional tobacco and smoke-free tobacco/nicotine products and was categorized as: (1) non-smoker; and (2) smoker (referring to daily or occasional smoking/consumption of any tobacco/nicotine product). Physical activity was captured by the frequency of engaging in recreational sports activities requiring moderate physical effort (for a minimum of 30 min) and aggregated into a binary variable: (1) active ≥ 4 times a week (high activity); and (2) active ≤ 3 times a week (low activity). Self-assessed dietary habits were also dichotomized into a binary variable: (1) healthy (very or rather healthy); and (2) unhealthy (neither healthy nor unhealthy/rather or very unhealthy).

### 2.3. Analytic Sample and Statistical Analysis

The analytic sample for this study consisted of 1781 individuals (620 males and 1161 females) who responded in both baseline and follow-up waves (Table 1). Additional inclusion criteria were defined as having completed the EQ-5D-3L in both waves and having an overall item non-response < 10%.

HRQoL, measured by EQ-5D index scores, in the baseline and follow-up surveys was described by means and standard deviations (SD). Changes in HRQoL by socio-demographic and health behaviour indicators were presented as means within categories; by counts and proportions for increased and decreased EQ-5D index scores within categories of predictor variables. To indicate whether EQ-5D index score differences between variable categories were statistically significant, *t*-tests were used. Additionally, the prevalence of health problems by health domain in both measurements was presented to describe the change in health states.

Tobit regression modelling was used to study which variables had an impact on EQ-5D index scores in both baseline and follow-up surveys. As the distribution of EQ-5D index scores was positively skewed (best health status bounded to 1), Tobit models were considered suitable for such censored or bounded data [29]. Mutually adjusted models were built for baseline and follow-up measurements, with respective EQ-5D index scores serving as dependent variables and socio-demographic and behavioural indicators at baseline as predictor variables. The results were presented as beta coefficients along with 95% confidence intervals and *p*-values. The beta coefficient indicates the mean change in the reference value within the variable. The threshold for statistical significance was set at *p* < 0.05 throughout the analyses. All statistical analyses were performed using the STATA 14 software [30].

## 3. Results

The characteristics of the study sample (*n* = 1781) in both baseline and follow-up measurements are presented in Table 1. The mean age with standard deviation was 53.8 ± 17.6 years at the baseline. The majority of respondents were female (65.2%), Estonian (86.1%) and with higher education (53.0%). Excluding dietary habits, where the proportion of respondents reporting healthy diets increased from 43.7% to 51.1% (*p* < 0.05), no statistically significant changes were observed. The baseline and follow-up HRQoL by socio-demographic and health behaviour indicators are shown in Table 2. At the baseline, the mean EQ-5D index score for the study sample was 0.859 ± 0.177. The mean values of the EQ-5D index score varied significantly (*p* < 0.05) by age (18–29 vs. 55 and older), employment and financial status, physical activity and dietary habits. No statistically significant differences in HRQoL were found in sex, ethnicity, education, smoking or alcohol consumption.

In the follow-up data, the overall mean EQ-5D index score declined by 0.048 (*p* < 0.05) to 0.811 ± 0.188. Statistically significant declines in EQ-5D index score means were observed in all categories of the socio-demographic and behavioural variables considered. In addition, the existing socio-demographic health gradients from the baseline increased with statistically significant differences in HRQoL. The most notable declines were seen among respondents with primary or lower education (mean difference −0.072) and among those with financial difficulties (mean difference −0.062). In total, HRQoL increased for 13.6% and decreased for 39.4% of individuals. The largest decline in HRQoL was observed among respondents who consumed alcohol 4 or more times per week (50.8%), followed by those with primary or lower education (50.5%), unemployed or non-active individuals (43.7%) and those with financial difficulties (43.7%).

Changes in HRQoL during the pandemic can also be characterised by prevalence of reported health problems in health domains. No health problems in any of the five domains were reported by 48.0% (*n* = 855) of respondents at baseline and 33.4% (*n* = 595) at follow-up. The proportions (with 95% confidence intervals) of reported (some or severe) problems by EQ-5D domain at baseline and follow-up are presented in Figure 2. Although the proportion of reported problems rose in all health domains, the relative increase was the largest for anxiety/depression (baseline 27.7%; follow-up 42.5%) and pain/discomfort (baseline 39.5%; follow-up 50.0%).

The results of the regression analysis for association between HRQoL and socio-demographic and behavioural variables are presented in Table 3. Older age (≥55 vs. 18–29 years), unemployment or being economically non-active and having financial difficulties had a negative effect on HRQoL index scores at both baseline and follow-up measurements. Dietary habits were the only behavioural indicator associated with HRQoL at baseline.

At follow-up, differences in HRQoL by sex became statistically significant, with women having a lower HRQoL score compared to men. While HRQoL did not vary significantly by educational level in the baseline data, having higher education (compared to primary or lower education) proved to be a protective factor regarding HRQoL. In addition to dietary habits, low physical activity had a negative effect on the HRQoL score in the follow-up data.

## 4. Discussion

The results of this longitudinal study indicate that HRQoL among Estonian adults in the study population deteriorated substantially during the COVID-19 pandemic between June 2020 and April 2021. Statistically significant declines in mean EQ-5D index scores were observed for all socio-demographic and behavioural variables considered. Most of this was due to the increased reporting of problems in the pain/discomfort and anxiety/depression health domains. Mutually adjusted regression models revealed that older age, being unemployed/economically non-active and having financial difficulties were the socio-demographic factors significantly associated with lower HRQoL at both baseline and follow-up measurements. In the follow-up data, women had significantly lower HRQoL compared to men, whereas having tertiary or higher education proved to be the only protective factor regarding HRQoL. Unhealthy dietary habits and low physical activity had a negative impact on HRQoL scores in the follow-up data.

Before discussing these findings in detail, several potential limitations of the study and the data need to be addressed. First, the pandemic’s impact on HRQoL can only be partially explained by the current study as the baseline measurements were captured in the middle of the pandemic (although during the period of low infection rates). Second, the data represent a subset of respondents in the baseline survey. While the baseline survey had 3464 respondents in total, only 1781 (51%) met the inclusion criteria. Due to attrition rate and non-response bias (e.g., 62.4% of respondents were female), the results cannot be generalized directly to the whole population; however, data within subgroups are representative. The third methodological consideration relates to the survey mode. As a self-administered online questionnaire was the only feasible option for a rapid assessment survey, the nationally representative sample (by sex and age distribution) was only able to include individuals with valid e-mail addresses in the population registry database. However, as an earlier study [31] has shown that approximately 90% of individuals have a valid e-mail address in the population registry database, we do not consider the potential selection bias during sampling a serious threat to the representational quality of our data. Fourth, while the analytical sample included individual level follow-up data, the causality per se for both HRQoL and behavioural changes cannot be determined as the specific timing and context of individual changes are not covered in the data. Fifth, the response bias arising from the self-administered questionnaire should be considered. In order to reduce the bias across socio-demographic variables, we validated sex and age information from Estonian national registry data, but information for the ethnicity and education variables is drawn from the baseline survey. Despite these considerations, the main strength of the study is the use of longitudinal data from a survey purposefully designed and timed to assess the impact of the COVID-19 pandemic on the population’s health.

Our results correspond to findings from previous longitudinal surveys from Den-mark [19] and Japan [20], which reported a decline in HRQoL during the pandemic, although these studies did not use the EQ-5D for assessing HRQoL. At the baseline (in June 2020), the mean EQ-5D index score for the Estonian sample (0.859) was similar to that of the Moroccan general population (0.86) during home confinement in May 2020 [32] but lower than that among the Chinese general population (0.949) in March 2020 [33].

Although the proportion of reported problems rose in all health domains, the decline in HRQoL can mostly be explained by the pain/discomfort and anxiety/depression health domains, as their relative increase between two measurements was larger compared to other domains. Comparable results reporting an increase of problems in the pain/discomfort and anxiety/depression health domains were found among the Chinese population in March 2020 [34]. Compared to a Moroccan study [32] where mobility, self-care, usual activities and pain/discomfort were most affected during the confinement compared to pre-pandemic reported levels, the relative importance of the mental health domain in our data is noteworthy. The relative importance of mental health problems in HRQoL is supported by findings from studies in Vietnam [35] and China [34].

In this study, the mean values of HRQoL score varied significantly by age, sex, education, employment status and financial situation of respondents. However, these socio-demographic patterns changed when baseline and follow-up data were compared. At the baseline, HRQoL differed significantly by age, employment and financial status, whereas additional sex and educational differences emerged in the follow-up. The age variation in the HRQoL where the mean EQ-5D index score decreased with increasing age was largely expected and has been reported in several other studies covering the pandemic period [15,33,36]. The lower HRQoL of women compared to men in the follow-up data is also in accordance with previous studies [15,32,36,37]; however, given that there were no differences in sex in the baseline data, the pandemic had a stronger negative impact on women’s health assessment compared to men. Similar effects were also found for education, where the relative differences between educational levels increased during the follow-up. These results correspond to earlier studies [32,38] that have found higher education to be associated with better HRQoL. Moreover, the most notable declines in HRQoL were seen in respondents with primary or lower education, whereas having tertiary or higher education proved to be a protective factor regarding HRQoL scores in the regression analysis. Differences in HRQoL by employment and financial status were present in both baseline and follow-up data. Unemployed or economically non-active respondents and those with financial difficulties had substantial declines in mean HRQoL values. A similar result was found in another study [36] where employed respondents had higher HRQoL scores compared to those not employed. As primary or lower education level, being economically non-active and having financial difficulties were all statistically significantly associated with lower HRQoL in the follow-up data, it is plausible that the pandemic has had a disproportionally higher impact on those with a lower socio-economic status.

Unhealthy dietary habits were the only behavioural indicator that had a significant effect on HRQoL in both baseline and follow-up surveys. The association remained significant in an additional analysis focusing on the change in health behaviour indicators during the pandemic. This could partly be explained by the possible change in eating patterns and meal preparation during the confinement and the wider application of remote work and home-schooling, which has been noted elsewhere as well [32,38,39]. Thus, the positive association between HRQoL and healthy dietary habits is expected and has been demonstrated by an earlier study during the pandemic [40].

No statistically significant differences in HRQoL were found for smoking or alcohol consumption. However, the mean HRQoL score was lower among high-risk alcohol consumers compared with low-risk consumers. In addition, the largest proportion of respondents with a decline in HRQoL score were among high-risk alcohol consumers. A previous study among suspected COVID-19 patients showed that the HRQoL score was significantly higher in people who did not consume alcohol [37]. In line with a previous study [37], our findings showed that respondents with high physical activity during the pandemic had higher HRQoL scores than individuals with a low physical activity level. A similar result was seen in a study during the pandemic when HRQoL was measured using the SF-8 [34].

## 5. Conclusions

During the COVID-19 pandemic, HRQoL declined substantially among the study population of Estonian adults. The difference in EQ-5D-3L mean scores between the baseline and the follow-up was significant within all demographic, socio-economic and health behaviour categories considered. Two EQ-5D domains that affected HRQoL decline were pain/discomfort and anxiety/depression. Moreover, those with lower education or financial difficulties seemed to be most affected, whereas the impact of health behaviour on HRQoL during the pandemic was relatively limited. Thus, the study demonstrated that the COVID-19 pandemic had an overall negative effect on HRQoL and its impact was stronger in vulnerable population groups. Age, dietary habits and employment and financial status had a statistically significant effect on HRQoL at baseline, and sex, education and physical activity additionally at follow-up. Consequently, the relative health inequalities in general likely increased during and as a result of the pandemic. Further longitudinal studies with extended follow-up periods are needed to assess whether these socio-demographic health discrepancies are present in other countries within the European region and whether these changes affect individual health and its trajectories for longer periods.

## Figures and Tables

**Figure 1 ijerph-19-09060-f001:**
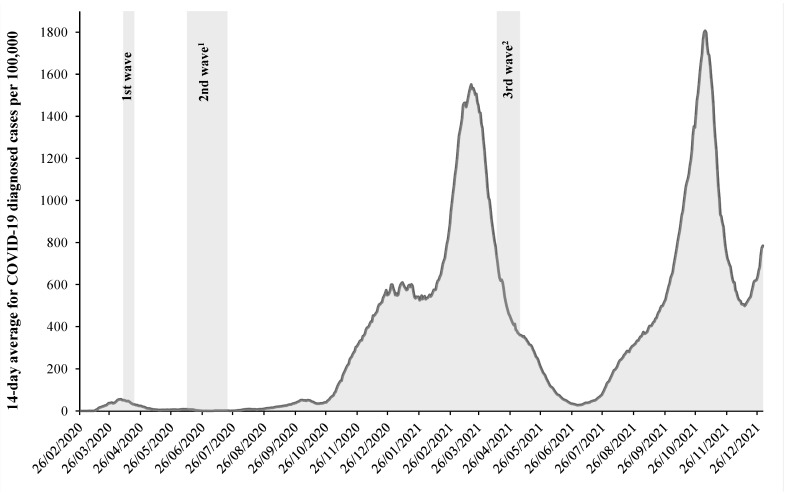
Survey periods and 14-day average incidence rates for COVID-19 in Estonia, 2020–2021 [24]. ^1^ HRQoL baseline measuring; ^2^ HRQoL follow-up measuring.

**Figure 2 ijerph-19-09060-f002:**
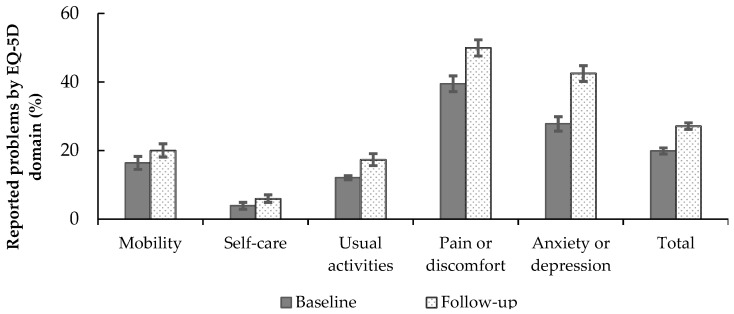
Proportion (%) with 95% confidence intervals of reported problems by EQ-5D domains at baseline and follow-up.

**Table 1 ijerph-19-09060-t001:** Demographic, socio-economic and health behaviour characteristics of study sample at baseline and follow-up, Estonia 2020–2021.

Variables		Baseline ^1^		Follow-Up	
		*n*	% ^2^	*n*	% ^2^
**Sex**	Male	620	34.8	-	-
	Female	1161	65.2	-	-
**Age, years**	18–29	210	11.8	-	-
	30–44	363	20.4	-	-
	45–54	273	15.3	-	-
	55–64	307	17.2	-	-
	≥65	628	35.3	-	-
**Ethnicity**	Estonian	1534	86.1	-	-
	Other	247	13.9	-	-
**Education**	Primary or lower	110	6.2	-	-
	Secondary/vocational	727	40.8	-	-
	Tertiary/higher	944	53.0	-	-
**Employment status**	Employed	1120	62.9	1134	63.7
	Unemployed/non-active	659	37.0	645	36.2
	*Missing*	2	0.1	2	0.1
**Financial status**	Comfortable	469	26.3	512	28.8
	Sufficient	1076	60.4	1053	59.1
	Difficulty coping	229	12.9	212	11.9
	*Missing*	7	0.4	4	0.2
**Alcohol consumption**	≤3 times per week	1687	94.7	1686	94.7
	≥4 times per week	59	3.3	88	4.9
	*Missing*	35	2.0	7	0.4
**Smoking status**	Non-smoker	1444	81.1	1446	81.2
	Smoker	330	18.5	322	18.1
	*Missing*	7	0.4	13	0.7
**Physical activity**	≥4 times per week	450	25.3	455	25.6
	≤3 times per week	1288	72.3	1304	73.2
	*Missing*	43	2.4	22	1.2
**Dietary habits**	Healthy	778	43.7	910	51.1
	Unhealthy	989	55.5	867	48.7
	*Missing*	14	0.8	4	0.2

^1^ for sex, age, ethnicity and education indicators, the baseline data were used; sex and age were controlled by national registry data. ^2^ proportions were calculated from total subsample (*n* = 1781).

**Table 2 ijerph-19-09060-t002:** EQ-5D-3L index scores at baseline and follow-up by sample characteristic and changes in HRQoL, Estonia 2020–2021.

Variables		Baseline			Follow-Up		Change in HRQoL		
		*n*	Mean	SD	Mean	SD	Difference in Means ^1^	Increase, *n* (%)	Decrease, *n* (%)
**Total sample**		1781	0.859	0.177	0.811	0.188	0.048 *	242 (13.6)	701 (39.4)
**Sex**	Male	620	0.860 ^ref^	0.173	0.821 ^ref^	0.175	0.039 *	86 (13.9)	221 (35.6)
	Female	1161	0.858	0.180	0.805 *	0.195	0.053 *	156 (13.4)	480 (41.3)
**Age, years**	18–29	210	0.902 ^ref^	0.156	0.858 ^ref^	0.167	0.044 *	30 (14.3)	82 (39.0)
	30–44	363	0.912	0.151	0.856	0.172	0.055 *	34 (9.4)	141 (38.8)
	45–54	273	0.897	0.163	0.845	0.188	0.053 *	32 (11.7)	101 (37.0)
	55–64	307	0.838 *	0.183	0.789 *	0.199	0.049 *	32 (10.4)	118 (38.4)
	≥65	628	0.806 *	0.185	0.764 *	0.187	0.042 *	99 (15.8)	259 (41.2)
**Ethnicity**	Estonian	1534	0.858 ^ref^	0.173	0.810 ^ref^	0.188	0.048 *	212 (13.8)	606 (39.5)
	Other	247	0.860	0.205	0.816	0.191	0.044 *	30 (12.1)	95 (38.5)
**Education**	Primary or lower	110	0.839 ^ref^	0.207	0.767 ^ref^	0.203	0.072 *	15 (13.6)	55 (50.0)
	Secondary/vocational	727	0.846	0.190	0.796 *	0.202	0.050 *	105 (14.4)	302 (41.5)
	Tertiary/higher	944	0.870	0.162	0.827 *	0.174	0.043 *	122 (12.9)	344 (36.4)
**Employment status**	Employed	1120	0.891 ^ref^	0.144	0.846 ^ref^	0.154	0.046 *	144 (12.9)	413 (36.9)
	Unemployed/non-active	659	0.803 *	0.212	0.747 *	0.223	0.051 *	98 (14.9)	288 (43.7)
**Financial status**	Comfortable	469	0.913 ^ref^	0.127	0.876 ^ref^	0.144	0.035 *	55 (11.7)	157 (33.5)
	Sufficient	1076	0.859 *	0.159	0.804 *	0.173	0.052 *	150 (13.9)	444 (41.3)
	Difficulty coping	229	0.748 *	0.267	0.686 *	0.271	0.062 *	36 (15.7)	100 (43.7)
**Alcohol consumption**	≤3 times/week	1687	0.860 ^ref^	0.174	0.812 ^ref^	0.186	0.048 *	227 (13.5)	668 (39.6)
	≥4 times/week	59	0.841	0.215	0.799	0.233	0.042 *	14 (23.7)	30 (50.8)
**Smoking status**	Non-smoker	1444	0.861 ^ref^	0.171	0.814 ^ref^	0.182	0.045 *	199 (13.8)	552 (38.2)
	Smoker	330	0.849	0.203	0.799	0.215	0.056 *	41 (12.4)	139 (42.1)
**Physical activity**	≥4 times/week	450	0.875 ^ref^	0.157	0.848 ^ref^	0.158	0.035 *	57 (12.7)	155 (34.4)
	≤3 times/week	1288	0.855 *	0.182	0.797 *	0.197	0.053 *	180 (14.0)	541 (42.0)
**Dietary habits**	Healthy	778	0.884 ^ref^	0.153	0.836 ^ref^	0.176	0.045 *	108 (13.9)	313 (40.2)
	Unhealthy	989	0.827 *	0.200	0.784 *	0.198	0.051 *	133 (13.4)	386 (39.0)

^1^ difference between baseline and follow-up data of EQ-5D-3L index score means, paired *t*-test used; ^ref^ reference category for within variable differences in HRQoL means, *t*-test used; * statistically significant difference at *p* < 0.05.

**Table 3 ijerph-19-09060-t003:** Relationship between EQ-5D-3L index score and demographic, socio-economic and health behaviour indicators at baseline and follow-up, Estonia 2020–2021.

Variables		Baseline		Follow-Up	
		Beta (95% CI)	*p*-Value	Beta (95% CI)	*p*-Value
**Sex**	Male	ref		ref	
	Female	−0.017 (−0.047; −0.014)	0.275	−0.033 (−0.059; −0.008)	0.010
**Age, years**	18–29	ref		ref	
	30–44	−0.013 (−0.069; 0.043)	0.643	−0.020 (−0.066; 0.025)	0.379
	45–54	−0.026 (−0.086; 0.033)	0.382	−0.028 (−0.077; 0.020)	0.252
	55–64	−0.139 (−0.196; −0.082)	<0.001	−0.102 (−0.148; −0.055)	<0.001
	≥65	−0.162 (−0.212; −0.111)	<0.001	−0.107 (−0.149; −0.655)	<0.001
**Ethnicity**	Estonian	ref		ref	
	Other	0.020 (−0.022; 0.063)	0.339	0.018 (−0.016; 0.054)	0.300
**Education**	Primary or lower	ref		ref	
	Secondary/vocational	0.011 (−0.051; 0.073)	0.722	0.038 (−0.013; 0.090)	0.140
	Tertiary/higher	0.019 (−0.042; 0.082)	0.537	0.048 (0.003; 0.099)	0.067
**Employment status**	Employed	ref		ref	
	Unemployed/ non-active	−0.081 (−0.116; −0.045)	<0.001	−0.087 (−0.116; −0.057)	<0.001
**Financial status**	Comfortable	ref		ref	
	Sufficient	−0.054 (−0.090; −0.018)	0.004	−0.066 (−0.095; −0.037)	<0.001
	Difficult to cope	−0.189 (−0.240; −0.137)	<0.001	−0.189 (−0.233; −0.146)	<0.001
**Alcohol consumption**	≤3 times/week	ref		ref	
	≥4 times/week	−0.049 (−0.127; 0.029)	0.217	−0.022 (−0.079; 0.034)	0.450
**Smoking status**	Non-smoker	ref		ref	
	Smoker	0.007 (−0.031; 0.045)	0.716	−0.002 (−0.034; 0.029)	0.881
**Physical activity**	≥4 times/week	ref		ref	
	≤3 times/week	−0.017 (−0.050; 0.017)	0.324	−0.049 (−0.077; −0.021)	0.001
**Dietary habits**	Healthy	ref		ref	
	Unhealthy	−0.085 (−0.115; −0.056)	<0.001	−0.058 (−0.083; −0.033)	<0.001

## Data Availability

Data can be made available upon request to the corresponding author.

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
