# Peer review of "Health-Related Quality of Life and Its Socio-Demographic and Behavioural Correlates during the COVID-19 Pandemic in Estonia"

_ijerph, 2022, doi:10.3390/ijerph19159060_

Round 1

Reviewer 1 Report

The study focused on analyzing the changes in health-related quality of life among Estonian general population with longitudinal data of 1781 participants. Although interesting to see such studies, I feel there was a big problem of the analysis. The data was not sufficiently analyzed and compared with other nations in general. What would be the most important thing of choosing Estonian to study? What specific reason do they have? Also, I am very concerned on the data analysis because I would assume they use Cross-lagged panel model to analyze their data.

Author Response

Dear Reviewer 1, 

Thank you very much for all your comments and questions. We revised our manuscript based on three reviews and edited the English language. Additions and changes are highlighted in the text of the manuscript. Hereby I am also answering your specific comments and questions. 

Point 1: “The data was not sufficiently analyzed and compared with other nations in general. What would be the most important thing of choosing Estonian to study? What specific reason do they have?” 

Response 1: Health-related quality of life (HRQoL) among the general population, especially in Europe, is not much studied compared to clinical samples or measured with the EQ-5D descriptive system during the COVID-19 pandemic. There is as well a lack of analyses based on longitudinal data that help to understand how the pandemic affected HRQoL among general populations. Therefore, it was complicated to bring comparisons with other nations in line with our results, however, one additional reference was added (page 9 line 281). 

Estonia is a small European country with 1.3 million inhabitants that we consider rather a strength as it is possible to conduct nationally representative studies that might be harder in more populated countries. The Estonian rapid-assessment survey on Covid was possible to conduct in a short time frame when the follow-up survey coincided with the period where Estonian 14-day COVID-19 infection rates (>1500 cases per 100,000 in the 11th week of 2021) were the highest in the world.  

Point 2: “Also, I am very concerned on the data analysis because I would assume they use Cross-lagged panel model to analyze their data.”

Response 2: Thank you for this methodological suggestion. We took it into consideration and became more aware of the imprecise use of the English language and previously insufficient results interpretation via the Tobit regression model. Therefore, we revised the methodology part and concentrated on adding more information to the survey (page 2 lines 79-82, lines 86, 98, 109-119) as well as corrected the use of language by aim and research questions (page 1 lines 10, page 2 lines 59,68,71,73). Corrections related to analysis are done on page 1 line 14, page 5 lines 166,168-169, 172-173. 

Sincerely,

Merili Tamson (corresponding author)

National Institute for Health Development

Tel: +37256807737

Reviewer 2 Report

I was invited to revise the paper entitled "Health-related quality of life and its socio-demographic and behavioural correlates during the COVID-19 pandemic in Estonia". It was a longitudinal study that aimed to evaluate the changes in health-related quality of life (HRQoL) among Estonian general population and its socio-demographic and behavioural correlates during COVID-19 pandemic.

I have to say that this is a very well written manuscript and it is a nice study of this topic. The standard of academic writing is rich, the bibliographic references are very up-to-date and the proposal statistics analyses are according the objectives and are perfect.

Just a brief observation: The authors referred that "More details on the survey can be found elsewhere (24)”.  My only criticism is that the document is not in English and I cannot read to being well informed about the subject.

Author Response

Dear Reviewer 2, 

Thank you very much for your positive review and comments. We revised our manuscript based on three reviews and edited the English language. Additions and changes are highlighted in the text of the manuscript. Hereby I am also answering your specific comment. 

Point 1: “The authors referred that "More details on the survey can be found elsewhere (24)”.  My only criticism is that the document is not in English and I cannot read to being well informed about the subject.” 

Response 1: The report of the rapid assessment survey on COVID is written in Estonian and has a summary in English. Therefore, we added an additional explanation about the survey (page 2 lines 79-82, 89-94, 98, 100-110) and specified survey periods in Figure 1 (page 3). 

Sincerely,

Merili Tamson (corresponding author)

National Institute for Health Development

Tel: +37256807737

Reviewer 3 Report

The authors conducted a study to examine changes in HRQoL by COVID-19 pandemic, associations between attributes and HRQoL, and attributes and changes during the course of the pandemic.

Although there have been previous studies on mental illness, suicide, and other illnesses, few studies have used changes in HRQoL as an outcome. As the authors used stratified random sampling from a population registry database to collect data,  consideration has been given to the selection of the target population, although there is selection bias due to the limited number of respondents as described by the authors.

However, there are several issues with this paper.

1.       one of the authors' hypotheses is to examine the association between individual attributes and changes in HRQoL during the pandemic. If that is the case, an outcome of the regression analysis should be the amount of change in HRQoL.

2.       The baseline for this study, June 2020, is already in the middle of the pandemic. In the setting of this study, changes due to the pandemic can only be partially explained. This point should also be mentioned. Also, as supporting information, it would be desirable to mention policies and timing, such as the lockdown in Estonia. Moreover, if there are references regarding changes in HRQoL prior to the pandemic, please mention any trends that would have existed independent of the pandemic.

3.       HRQoL was not asked in the first cross-sectional survey, only from the second survey. The flowchart should clearly indicate this situation.

4.       Table 1 compares baseline and follow-up attributes, but why does it not include information such as gender and age among follow-up respondents? The impact of loss-to-follow-up on the findings also does not seem to have been considered.

5.       I tried to reference this survey in detail, but could not find reference 24.

Author Response

Dear Reviewer 3, 

Thank you very much for your review. We revised our manuscript based on three reviews and edited the English language. Additions and changes are highlighted in the text. Hereby I am also answering your specific comments and questions.

Point 1: "As the authors used stratified random sampling from a population registry database to collect data, consideration has been given to the selection of the target population, although there is selection bias due to the limited number of respondents as described by the authors.”

Response 1: To improve the manuscript on this topic, we added additional information about the respondents into the methodology (page 3 lines 109-119).

Point 2: “One of the authors' hypotheses is to examine the association between individual attributes and changes in HRQoL during the pandemic. If that is the case, an outcome of the regression analysis should be the amount of change in HRQoL.”

Response 2: Thank you for bringing out imprecise use of language and results interpretation. The authors originally did not want to study the amount of change in HRQoL to be an outcome of the analysis. We modified the methodology part of the manuscript incl. the aim of the study and research questions (page 1 lines 10, 14; page 2 lines 60,68,71,73; page 5 lines 168-169, 172-173).

Point 3: “The baseline for this study, June 2020, is already in the middle of the pandemic. In the setting of this study, changes due to the pandemic can only be partially explained. This point should also be mentioned. Also, as supporting information, it would be desirable to mention policies and timing, such as the lockdown in Estonia. Moreover, if there are references regarding changes in HRQoL prior to the pandemic, please mention any trends that would have existed independent of the pandemic.”

Response 3: Thank you for this comment. We did add an extra point to the limitations part (page 8 lines 251-252) and supporting information about policies during the time of survey waves (page 3 lines 100-108).

In this analysis, we found the trend of HRQoL during the pandemic that was declining. However, no previous study of HRQoL in Estonia has been done among the general population and therefore it was not possible to evaluate the change of HRQol prior to the pandemic. One analysis has been in 2009 published in Estonian that indicated the EQ-5D score (0.59) among women with menopause [1] that is not relevant to compare in our discussion.

[1] Fischer, K.; et al. Postmenopausis naiste tervise enesehinnangu ja elukvaliteedi seos [Relationship between subjective well-being and quality of life in postmenopausal women]. Celsius Healthcare OÜ: Tartu, Estonia; 2009. https://eestiarst.ee/en/relationship-between-subjective-well-being-and-quality-of-life-in-postmenopausal-women/ 

Point 4: “HRQoL was not asked in the first cross-sectional survey, only from the second survey. The flowchart should clearly indicate this situation.”

Response 4: Corresponding changes were made in figure 3 (page 3).

Point 5: “Table 1 compares baseline and follow-up attributes, but why does it not include information such as gender and age among follow-up respondents? The impact of loss-to-follow-up on the findings also does not seem to have been considered.”

Response 5: Data from the self-administered online questionnaire include the risk for response bias that we also noticed. To reduce the data difference across the main socio-demographic variables, we decided to use fixed baseline information for this study analysis. Variables sex and age were controlled from the Estonian national registry and considered validated variables in the study. Information about ethnicity in the registry-based data was not complete and therefore data was not validated. The education variable was not included in the registry data and only self-completed questionnaire data was used. We added specific information to Table 1 footnote (page 5 lines 162-163) and the response bias to the limitation part as well (page 8, lines 266–270). 

The impact of loss-to-follow-up was not considered as the analyzed sample was the same in both survey waves. We included in the analysis only these respondents who did answer HRQoL questions in both survey waves. Those, who did not answer HRQoL questions were not included. We added more information about this into the methodological part (page 3 lines 109-119). Also, we did an additional analysis to take into consideration a loss to follow-up respondents in two cohorts. Therefore we compared the excluded sample results with the current analysis. Both models were used to control if models predict data correctly. The baseline model RMSE was 0.82, which indicates that the model was not suitable for all individuals. The follow-up model RMSE was 0.15, which showed that the model worked well for the data and results did not change for those who did not answer in the baseline but answered in follow-up.

Point 6: “I tried to reference this survey in detail, but could not find reference 24.”

Response 6: The web page for the survey report is now added to the reference (page 11 lines 395-396), however, only the summary is in English. Additional survey methodology information is also added in the manuscript (page 2 lines 77-82, 86; page 3 figure 1 and line 98).

Sincerely,

Merili Tamson (corresponding author)

National Institute for Health Development

Tel: +37256807737

Round 2

Reviewer 1 Report

Although the author relplied on my comments, I still think this study should not be accepted for publication because their reason for doing this study was because of the lack of data, which is not a scientific reason. Also, the data is longitudinal, but their analysis was mainly cross-sectional. So I don't feel this study could be revised because of such big issues.

Author Response

Dear Reviewer 1,

Thank you for your comments. Below you may find our response to your Round 2 review.

Point 1: Although the author relplied on my comments, I still think this study should not be accepted for publication because their reason for doing this study was because of the lack of data, which is not a scientific reason. Also, the data is longitudinal, but their analysis was mainly cross-sectional. So I don't feel this study could be revised because of such big issues.

Response 1:

In the authors opinion, a selected methodology (Tobit regression) is suitable for answering the second research question (b). We would note that the objectives need to be distinguished by the original rapid-assessment survey and the current study. The objectives are different, but lack of data is one of the main reasons for the rapid-assessment survey as well as the current article. As a side note, individual health trajectories during the COVID-19 pandemic will be analysed in a separate paper.

For improving our last version of manuscript, minor changes and additions in methodological part (page 2 line 71; page 3 lines 115-116), results interpretation (page 7 lines 211, 215-216), discussion (page 8 lines 243-244, 257-259, 262) and conclusions (page 10 lines 339-340, 353-354) has been done (page 8 lines 243-244, 262; page 10 lines 339-340, 353-354).

Sincerely,

Merili Tamson (corresponding author)

National Institute for Health Development

Tel: +37256807737

Reviewer 3 Report

I respect the authors' efforts to revise the paper.

Although the study remains strongly problematic for generalization, it is worthy of publication because it examines the impact of each attribute on HRQoL changes during the COVID-19 pandemic.

Author Response

Dear Reviewer 3,

Thank you for your comments. Below you may find our response to your Round 2 review.

Point 1: „Although the study remains strongly problematic for generalization, it is worthy of publication because it examines the impact of each attribute on HRQoL changes during the COVID-19 pandemic.“

Response 1:

Related to generalizability, we would like to add that the results of the our manuscript are highly relevant for the current/ongoing national initiatives for improving population mental health as they provide evidence on the importance of mental health domain (and its deterioration during the pandemic) for health status. Also, publishing findings from this population-based longitudinal survey are also relevant for the international research community as well because the available evidence on population-level HRQoL changes is limited. This is even more true for longitudinal data, which is one of the strengths of the current study (discussed in Introduction and Discussion sections).

For improving our last version of manuscript according to your comments, minor changes and additions has been done on the page 8 lines 243-244, 262; page 10 lines 339-340, 353-354.

Sincerely,

Merili Tamson (corresponding author)

National Institute for Health Development

Tel: +37256807737
